# The bilirubin-to-albumin ratio as a potential prognostic biomarker for all-cause mortality in patients with acute decompensated cirrhosis: A prospective study

Hien Viet Quang Nguyen[1‡], Thang Viet Luong[2,3‡], Trung Nguyen Tran[4,5],
Hai Nguyen Ngoc Dang[6]*

1 Hue Central Hospital, Hue, Vietnam, 2 University of Medicine and Pharmacy, Hue University, Hue, Vietnam, 3 Menzies Institute for Medical Research, University of Tasmania, Hobart, Tasmania, Australia, 4 NTT Hi-Tech Institute, Nguyen Tat Thanh University, Ho Chi Minh City, Vietnam, 5 Nguyen Tat Thanh University Center for Hi-Tech Development, Saigon Hi-Tech Park, Ho Chi Minh City, Vietnam, 6 Faculty of Medicine, Duy Tan University, Da Nang, Vietnam

‡ HNVQ and TVL are co-first authors and contributed equally to this work.
* ngochai123dc@gmail.com

## Abstract

### Background

Acute decompensation (AD) of liver cirrhosis is a life-threatening condition with high mortality and a significant healthcare burden. Although various prognostic models are available, their complexity often limits their practical utility in clinical settings. The bilirubin-to-albumin (B/A) ratio has emerged as a potential biomarker for critically ill patients. Despite its simplicity, the prognostic value of the B/A ratio in patients with AD remains uncertain.

### Methods

This two-center, prospective observational study screened 748 participants. After follow-up and application of the exclusion criteria, 279 patients with AD were included in the final analysis. The B/A ratio was evaluated for its association with 30-day, 90-day, and 180-day mortality.

### Results

Kaplan-Meier survival analysis demonstrated worse survival outcomes in patients with higher B/A ratios, with 180-day survival rates of 95.7%, 87.1%, 64.3%, and 56.5% from the lowest to highest quartiles, respectively. ROC curve analysis further validated its prognostic value, identifying optimal B/A ratio cutoffs of 3.30, 3.17, and 3.10 for predicting 30-day, 90-day, and 180-day mortality, with corresponding AUC values of 0.77, 0.79, and 0.75, indicating moderate predictive ability. The sensitivity and specificity at these cutoff points were 81.8% and 63.0% (30-day), 81.8% and

**Data availability statement:** All relevant data are within the manuscript and its Supporting information files.

**Funding:** The author(s) received no specific funding for this work.

**Competing interests:** The authors have declared that no competing interests exist.

66.0% (90-day), and 77.9% and 68.7% (180-day). Furthermore, the B/A ratio demonstrated a prognostic performance comparable to MELD, MELD-Na, MELD 3.0, Child–Pugh, ALBI, EZ-ALBI, and PALBI, with no statistically significant differences in AUC values ($p > 0.05$).

## Conclusion

The B/A ratio is a simple and effective prognostic biomarker, with a higher B/A ratio associated with increased mortality in patients with AD.

---

## Introduction

Acute decompensation (AD) of liver cirrhosis is a heterogeneous syndrome characteristic of end-stage liver disease, with a high mortality rate [1–3]. In addition to their high mortality rates, AD imposes substantial economic burdens on healthcare systems and society because of the significant costs associated with their management [4,5]. Early prognostic assessment is therefore essential to guide clinical decision-making and optimize treatment strategies. To address this need, several prognostic scoring systems have been developed [6]. However, the complexity of these models, which have multiple parameters, may limit their real-world applicability and delay their integration into clinical practice.

Among the various biomarkers associated with AD prognosis, bilirubin has been extensively studied. A strong relationship between elevated bilirubin levels and increased mortality risk has been well documented in both experimental and clinical research, particularly in acute hepatitis-related mortality [7,8]. Moreover, the level of albumin, a key hepatic synthetic protein, is an essential marker of liver function, and hypoalbuminemia is a direct consequence of hepatic dysfunction [9]. The interplay between bilirubin and albumin underpins the development of composite indices such as the albumin–bilirubin (ALBI) score and the platelet–albumin–bilirubin (PALBI) score, both of which have demonstrated prognostic value in chronic liver disease [10–15]. Nevertheless, these scores involve complex calculations, which may be impractical in urgent clinical settings. This raises the question of whether a simpler parameter, such as the bilirubin-to-albumin (B/A) ratio, might provide a practical and clinically meaningful alternative for mortality prediction in patients with liver disease.

Interestingly, recent studies have explored the B/A ratio as a potential biomarker for assessing brain injury, particularly in neonatal and pediatric populations [16–18]. In neonates, especially preterm infants, the underdeveloped blood–brain barrier allows unconjugated bilirubin to enter the central nervous system more readily, increasing the risk of bilirubin-induced neurotoxicity [19]. In contrast, under normal physiological conditions in adults, bilirubin and albumin do not easily cross the blood–brain barrier [20]. This paradox raises concerns about the relevance of the B/A ratio in evaluating neurological injury in adults and whether its prognostic significance extends beyond neonatal settings.

Despite this theoretical limitation, emerging evidence suggests that the B/A ratio may still hold prognostic value in critically ill patients, independent of its role in neurotoxicity [21]. These findings indicate a potential association between the B/A ratio and AD outcomes, warranting further investigation. However, despite its simplicity and ease of calculation, the predictive value of the B/A ratio in adult patients with AD remains unclear, especially in comparison with well-established prognostic models and scores that have been externally validated.

## Methods

### Study design

We conducted a prospective study that integrated elements from the STROBE statement to enhance the reporting of observational research. The study received approval from the Ethics Council of Hue University of Medicine and Pharmacy (Approval number: H2022/291). This study was carried out in full compliance with the principles of the Declaration of Helsinki (2013 version).

### Study population

A convenience sampling method was employed in this study. To ensure methodological validity, the minimum sample size was calculated using a formula for descriptive studies based on a single population proportion, as no published data on all-cause mortality specifically among patients with AD of cirrhosis were available. Previous research involving over 90,000 patients with decompensated cirrhosis listed for liver transplantation reported an all-cause mortality rate of approximately 18% [22]. Another study indicated that mortality in patients with AD excluding ACLF was around 25% [23], while a different study suggested that patients with AD, including those with ACLF, may have a 180-day liver-related mortality rate approaching 35% [24]. Therefore, for sample size estimation, we conservatively selected the upper-bound mortality rate of 35%. Using a 95% confidence level (Z = 1.96) and a margin of error of 10%, the minimum required sample size was 88 patients. After adjusting for an anticipated 10% loss to follow-up, the final required sample size was determined to be 98 patients.

This study employed convenient sampling at two centers and included 748 participants. Of these, 279 participants who met the inclusion criteria and had complete follow-up data were included in the analysis. Hospitalized patients who were diagnosed and treated for AD at Hue Central Hospital and Hue University of Medicine and Pharmacy Hospital between 01/06/2023, and 01/08/2024, were included. AD was defined as any first or recurrent grade 2 or 3 ascites developing within less than 2 weeks, first or recurrent acute hepatic encephalopathy in patients with previously normal consciousness, acute gastrointestinal bleeding, or any type of acute bacterial infection [25]. The exclusion criteria included patients with incomplete medical records; missing data for 30, 90, or 180 days post treatment; hepatocellular carcinoma; severe comorbid chronic diseases outside the liver, such as advanced-stage chronic obstructive pulmonary disease, end-stage chronic kidney disease, end-stage heart failure, or advanced cancer; those using immunosuppressive drugs (except alcohol-related hepatitis patients using corticosteroids); and those infected with human immunodeficiency virus. The sampling and exclusion process is illustrated in Fig 1.

### Clinical data collection and outcomes

Patient history, previous medical conditions, and clinical examinations were performed.

The tests conducted in the study included a complete blood count, international normalized ratio (INR), creatinine, bilirubin, alanine aminotransferase (ALT), aspartate aminotransferase (AST), and albumin. Additionally, an abdominal ultrasound was performed to examine the liver, kidneys, and other abdominal organs for abnormalities. Furthermore, each patient had prognostic scores calculated, including the Child-Pugh score, Model for End-Stage Liver Disease (MELD), MELD-Na, MELD 3.0, ALBI, EZ-ALBI, and PALBI [14,15,26,27].

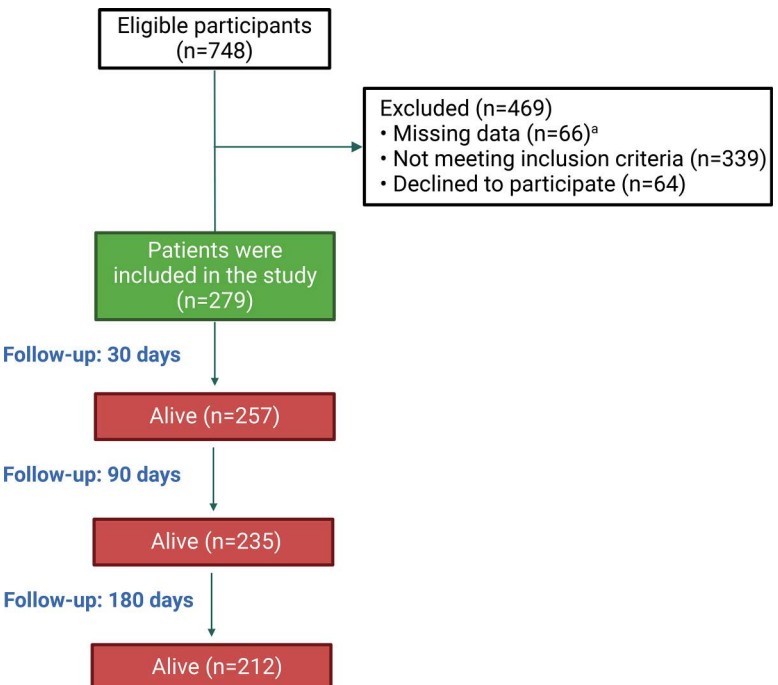

**Fig 1. Study flowchart of patient enrollment and follow-up.** [a]Missing data refer to cases where patients, prior to enrollment in the study, lacked complete information on medical history, laboratory results, or clinical background.

Patients were followed until death or day 180, and all-cause mortality at 30, 90, and 180 days was determined from hospital records and follow-up calls.

## Statistical analysis

The data were analyzed via SPSS version 26.0 (IBM Corp., Armonk, NY, USA) and R Studio version 4.4.1 (R Foundation for Statistical Computing, Vienna, Austria). The normality of the data distribution was assessed via the Shapiro–Wilk test. Continuous variables are expressed as the mean ± standard deviation for normally distributed data and as the median [interquartile range] for nonnormally distributed data. Categorical variables are presented as counts (%). The classification of the B/A ratio groups was based on the percentile distribution of all included patients. For comparisons among multiple groups, one-way ANOVA was performed for normally distributed data, whereas the Kruskal–Wallis test was used for nonnormally distributed data. Categorical variables were analyzed via Fisher's exact test to examine associations between groups. The associations between the B/A ratio and 30-day, 90-day, and 180-day mortality rates were examined via both univariate (crude) and multivariate (adjusted) Cox proportional hazard models, adjusting for potential confounders to determine their independent effects on survival outcomes. Kaplan–Meier survival analysis was utilized to visualize survival probabilities across B/A ratio groups, and the log-rank test was applied to compare survival distributions. Receiver operating characteristic (ROC) curve analysis was conducted on the entire study population to assess the discriminatory power of the B/A ratio in predicting mortality, with the area under the curve (AUC), 95% confidence intervals (CIs), and p values reported. The optimal B/A ratio cutoff was determined via Youden's index to maximize sensitivity and specificity. To compare differences in the AUC among the B/A ratio and other predictive models and scoring systems, the Hanley & McNeil method was applied. Statistical significance was defined as a two-tailed p value of <0.05.

## Results

### Baseline clinical characteristics

S1 Table presents a comparison of demographic characteristics, clinical features, and risk factors in the overall study population. The majority of participants were male (78.49%), with a high prevalence of alcohol use (71.33%). Regarding clinical presentation, jaundice was observed in 77.06% of patients, and ascites was present in 85.30%.

In terms of precipitating factors for hospitalization, acute viral hepatitis was the most common (36.2%), followed by alcohol abuse (29.0%) and unclear causes (26.5%). Hepatotoxic drug use accounted for 21.9%, while gastrointestinal bleeding and infections each contributed 18.6%. Surgical or procedural interventions were the least frequent triggers, accounting for only 1.8%. Additional details are provided in S1 Table and Fig 2.

### Comparison of the differences in demographic and laboratory characteristics between the B/A ratio groups by percentiles in patients with AD

Table 1 presents the demographic and laboratory characteristics across B/A ratio groups: G1 (<0.95), G2 (0.95–2.44), G3 (2.45–4.78), and G4 (≥4.79). Significant differences were observed in several key parameters, including WBC, NEU, total bilirubin, ALT, AST, albumin, sodium, and the INR. The WBC and NEU levels progressively increased from G1 to G4, with significant differences across the groups (p < 0.001). The total bilirubin, ALT, and AST levels also increased significantly across the groups (p < 0.001), whereas the albumin level decreased as the B/A ratio increased (p < 0.001). The INR significantly increased in the high B/A ratio groups (p < 0.001). Further details on all the variables are provided in Table 1.

### Results of adjusted and unadjusted Cox proportional hazard models in the study population

Table 2 presents the results of the crude and adjusted Cox proportional hazard models for 30-day, 90-day, and 180-day mortality. The B/A ratio was significantly associated with increased mortality risk across all time points in both the

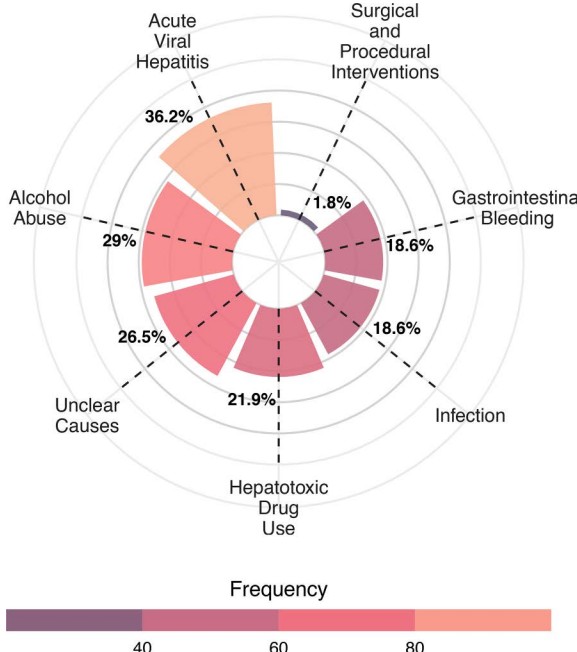

**Fig 2. Risk factors for hospitalization.**

**Table 1. Demographic and laboratory characteristics across B/A ratio groups by percentiles in patients with AD.**

| B/A ratio group | G1 (<0.95) | G2 (0.95-2.44) | G3 (2.45-4.78) | G4 (≥4.79) | P-value |
|---|---|---|---|---|---|
| Number (n) | 70 | 70 | 70 | 69 | |
| Age (years) | 55.59±11.38 | 54.67±9.77 | 56.70±9.57 | 53.00±12.99 | 0.242 |
| Male (%) | 52 (74.29) | 58 (82.86) | 60 (85.71) | 49 (71.01) | 0.113 |
| ACLF | 0 (0) | 0 (0) | 52 (74.3) | 63 (91.3) | **<0.001** |
| SBP (mmHg) | 120 (110 - 130) | 120 (110 - 130) | 120 (110 - 133) | 120 (110 - 130) | 0.757 |
| DBP (mmHg) | 70 (70 - 80) | 70 (70 - 80) | 75 (60 - 80) | 70 (60 - 80) | 0.662 |
| BMI (kg/m$^2$) | 20.86 (18.81 - 23.02) | 20.76 (19.25 - 22.87) | 21.63 (19.86 - 24.01) | 22.04 (19.91 - 24.08) | 0.167 |
| WBC (10$^9$/L) | 5.22 (4.18 - 7.09) | 5.97 (4.60 - 7.66) | 6.61 (4.98 - 9.66) | 7.80 (6.38 - 11.25) | **<0.001** |
| NEU (10$^9$/L) | 3.28 (2.27 - 4.42) | 3.90 (2.81 - 5.41) | 4.21 (2.92 - 6.66) | 5.15 (3.83 - 7.70) | **<0.001** |
| LYM (10$^9$/L) | 1.25 (0.80 - 1.75) | 1.13 (0.81 - 1.89) | 1.44 (0.82 - 2.25) | 1.54 (0.90 - 2.24) | 0.164 |
| HGB (g/dL) | 9.60±2.75 | 10.10±2.60 | 10.06±1.97 | 10.69±1.96 | 0.059 |
| HCT (%) | 30.10 (25.60 - 38.78) | 32.80 (27.15 - 37.45) | 28.15 (24.98 - 34.17) | 31.90 (28.30 - 35.50) | 0.075 |
| MCV (fL) | 86.0 (75.4 - 99.0) | 98.1 (90.4 - 108.0) | 99.3 (92.3 - 103.6) | 97.5 (90.0 - 100.4) | **<0.001** |
| RDW (%) | 15.9 (14.2 - 18.4) | 16.5 (14.3 - 18.6) | 15.6 (14.1 - 18.3) | 16.4 (14.8 - 19.4) | 0.126 |
| PLT (10$^9$/L) | 92.0 (64.8 - 124.1) | 96.5 (62.0 - 123.5) | 85.2 (63.0 - 124.3) | 109.0 (72.5 - 173.5) | 0.105 |
| Total bilirubin (μmol/L) | 17.9 (10.4 - 24.2) | 40.6 (33.2 - 48.9) | 89.9 (85.7 - 97.0) | 227.5 (144.1 - 381.2) | **<0.001** |
| ALT (U/L) | 35.9 (26.0 - 57.4) | 39.7 (28.6 - 54.5) | 39.7 (24.9 - 67.7) | 72.1 (42.6 - 183.1) | **<0.001** |
| AST (U/L) | 43.5 (33.0 - 66.7) | 71.1 (47.2 - 115.3) | 83.4 (56.1 - 146.5) | 135.5 (86.8 - 302.2) | **<0.001** |
| Creatinine (μmol/L) | 75.0 (61.3 - 95.0) | 68.0 (60.7 - 89.8) | 78.9 (62.0 - 120.9) | 79.0 (58.4 - 125.5) | 0.185 |
| Albumin (g/L) | 32.2 (29.9 - 35.7) | 27.8 (24.8 - 31.2) | 25.7 (22.1 - 28.5) | 25.0 (22.1 - 28.2) | **<0.001** |
| Sodium (mEq/L) | 138 (134 - 140) | 136 (133 - 139) | 134 (132 - 137) | 133 (131 - 135) | **<0.001** |
| Potassium (mEq/L) | 3.6 (3.4 - 3.9) | 3.6 (3.2 - 3.9) | 3.4 (3.2 - 4.1) | 3.6 (3.2 - 4.0) | 0.361 |
| INR | 1.28 (1.13 - 1.42) | 1.36 (1.19 - 1.60) | 1.70 (1.53 - 1.93) | 1.96 (1.65 - 2.39) | **<0.001** |
| MELD | 9.87 (8.29 - 11.8) | 13.97 (12.07 - 16.61) | 19.86 (18.15 - 22.75) | 24.89 (21.71 - 30.81) | **<0.001** |
| MELD-Na | 9.57 (7.11 - 14.03) | 14.83 (10.48 - 19.53) | 22.64 (19.53 - 25.91) | 27.38 (23.77 - 31.99) | **<0.001** |
| MELD 3.0 | 12.37 (9.97 - 13.91) | 16.09 (13.34 - 19.70) | 22.96 (20.73 - 25.37) | 28.69 (25.54 - 33.74) | **<0.001** |
| Child-Pugh | 7 (6 - 8) | 9 (8 - 10) | 11 (10 - 12) | 11 (10 - 13) | **<0.001** |

The values are presented as the means ± standard deviations or medians [I – III] as appropriate. Bold values indicate p ≤ 0.050. Abbreviations: ACLF, acute on chronic liver failure, ALT, alanine aminotransferase; AST, aspartate aminotransferase; B/A ratio, bilirubin-to-albumin ratio; HGB, hemoglobin; INR, international normalized ratio; LYM, lymphocyte; MCV, mean corpuscular volume; MELD, model for end-stage liver disease, NEU, neutrophil; PLT, platelet count; RDW, red cell distribution width; WBC, white blood cell count.

**Table 2. Cox proportional hazard model results for mortality based on the B/A ratio in the study cohort.**

| Variable | Crude model HR (95% CIs) | Model I HR (95% CIs) | Model II HR (95% CIs) |
|---|---|---|---|
| **30-day mortality** | 1.158 (1.098 - 1.220)* | 1.133 (1.050 - 1.223)* | 1.042 (0.975 - 1.115)[ns] |
| **90-day mortality** | 1.166 (1.119 - 1.214)* | 1.112 (1.050 - 1.177)* | 1.037 (0.978 - 1.100)[ns] |
| **180-day mortality** | 1.139 (1.098 - 1.181)* | 1.090 (1.034 - 1.149)* | 1.016 (0.963 - 1.159)[ns] |

*indicates p ≤ 0.001; [ns] indicates not significant. Crude model: No adjustment; Model I: Adjusted for age and sex, white blood cell count, neutrophil count, mean corpuscular volume, aspartate aminotransferase, alanine aminotransferase, international normalized ratio, and sodium; Model II: Fully adjusted for factors with significant differences in Table 1. Abbreviations: B/A, bilirubin to albumin ratio; CI, confidence interval; HR, hazard ratio.

unadjusted model and Model I (adjusted for age, sex, white blood cell count, neutrophil count, mean corpuscular volume, aspartate aminotransferase, alanine aminotransferase, international normalized ratio, and sodium), with all associations reaching statistical significance ($p \leq 0.001$). However, in Model II, which was fully adjusted for significant factors from Table 1 as well as the risk factors for hospitalization presented in Fig 2, the association was no longer statistically significant. Further details are provided in Table 2.

Fig 3 illustrates the nonlinear relationship between the B/A ratio and the hazard ratio for mortality. The analysis revealed that the hazard ratio for mortality increases sharply as the B/A ratio increases.

## Survival and predictive performance of the B/A ratio in patients with AD

Fig 4 shows the Kaplan–Meier survival curves for the total population, with significant differences in survival probabilities across the B/A ratio groups. As the B/A ratio increases, the survival probability decreases, indicating that higher B/A ratios are associated with poorer survival outcomes. Further details are provided in Fig 4.

Fig 5 presents the receiver operating characteristic (ROC) curves for predicting 30-day, 90-day, and 180-day mortality, with corresponding AUC values of 0.77, 0.79, and 0.75, respectively. The optimal B/A ratio cutoffs were 3.30, 3.17, and 3.10 for the 30-, 90-, and 180-day predictions, yielding sensitivities of 81.8%, 81.8%, and 77.9% and specificities of 63.0%, 66.0%, and 68.7%, respectively.

## Comparison of the prognostic value of the B/A ratio with other established prognostic models and scoring systems in patients with AD

Table 3 and Fig 6 shows that the p values for the AUC comparisons between the B/A ratio and established prognostic models (MELD, MELD-Na, MELD 3.0, and Child-Pugh) are mostly greater than 0.05, indicating no significant difference in predictive performance. Moreover, comparisons between the B/A ratio and previously proposed scoring systems incorporating bilirubin and albumin (ALBI, EZ-ALBI, and PALBI) also revealed no significant differences among these indices.

Detailed statistical information is provided in Table 3 and Fig 6.

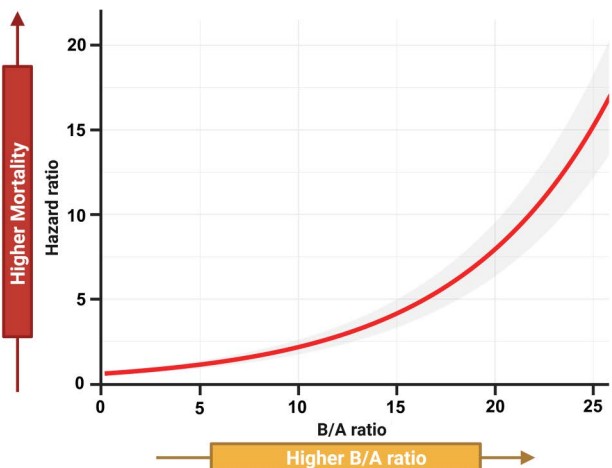

**Fig 3. Associations between the B/A ratio and mortality hazard ratio in patients with AD.**

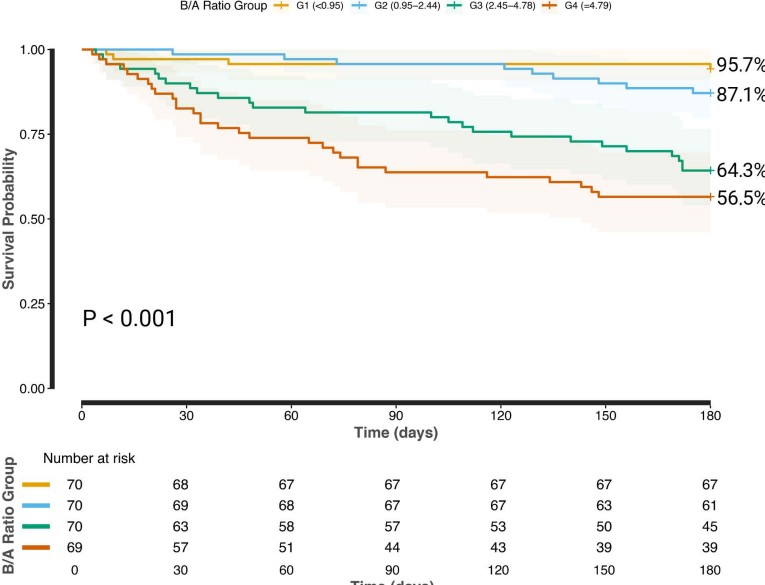

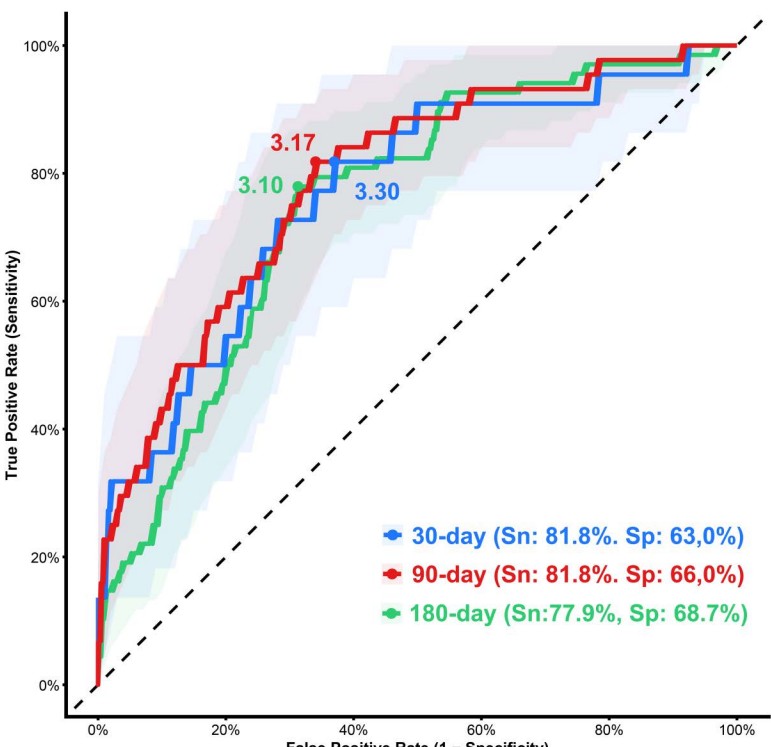

**Fig 4. Survival curves by B/A ratio groups in patients with AD.**

**Fig 5. ROC curves for 30-day, 90-day, and 180-day mortality prediction in patients with AD.**

**Table 3. Prognostic value of the B/A ratio and scoring systems for mortality in patients with AD at 30-day, 90-day, and 180-day time points.**

| Variables | AUC | 95% CI | | P value |
|---|---|---|---|---|
| **At the 30-day time point** | | | | |
| B/A ratio | 0.769 | 0.715 | 0.817 | <0.001 |
| Child-Pugh | 0.779 | 0.726 | 0.826 | <0.001 |
| MELD | 0.798 | 0.746 | 0.843 | <0.001 |
| MELD-Na | 0.811 | 0.760 | 0.855 | <0.001 |
| MELD 3.0 | 0.830 | 0.781 | 0.872 | <0.001 |
| ALBI | 0.808 | 0.716 | 0.900 | <0.001 |
| EZ-ALBI | 0.802 | 0.711 | 0.893 | <0.001 |
| PALBI | 0.761 | 0.661 | 0.861 | <0.001 |
| **At the 90-day time point** | | | | |
| B/A ratio | 0.789 | 0.736 | 0.835 | <0.001 |
| Child-Pugh | 0.771 | 0.717 | 0.819 | <0.001 |
| MELD | 0.800 | 0.748 | 0.845 | <0.001 |
| MELD-Na | 0.801 | 0.749 | 0.846 | <0.001 |
| MELD 3.0 | 0.825 | 0.775 | 0.868 | <0.001 |
| ALBI | 0.797 | 0.734 | 0.861 | <0.001 |
| EZ-ALBI | 0.809 | 0.744 | 0.873 | <0.001 |
| PALBI | 0.781 | 0.714 | 0.848 | <0.001 |
| **At the 180-day time point** | | | | |
| B/A ratio | 0.753 | 0.698 | 0.802 | <0.001 |
| Child-Pugh | 0.734 | 0.678 | 0.785 | <0.001 |
| MELD | 0.756 | 0.701 | 0.805 | <0.001 |
| MELD-Na | 0.746 | 0.691 | 0.796 | <0.001 |
| MELD 3.0 | 0.773 | 0.720 | 0.821 | <0.001 |
| ALBI | 0.754 | 0.692 | 0.817 | <0.001 |
| EZ-ALBI | 0.753 | 0.690 | 0.816 | <0.001 |
| PALBI | 0.760 | 0.699 | 0.822 | <0.001 |

Abbreviations: ALBI, albumin-bilirubin grade; B/A ratio, bilirubin-to-albumin ratio; EZ-ALBI, easy albumin-bilirubin grade; MELD, model for end-stage liver disease; MELD-Na, model for end-stage liver disease sodium; MELD 3.0, model for end-stage liver disease 3.0; PALBI, platelet-albumin-bilirubin grade.

## Discussion

The role of the B/A ratio in predicting the prognosis of patients with AD remains an area of ongoing investigation. In this study, we evaluated its prognostic value and found that higher B/A ratios were associated with poorer survival outcomes. A summarized graphical illustration of the study findings is presented in S1 Fig. Notably, the B/A ratio demonstrated comparable predictive performance to established prognostic models such as MELD, MELD-Na, and MELD 3.0. Furthermore, when compared with other scoring systems that incorporate bilirubin and albumin, including the ALBI grade, EZ-ALBI grade, and PALBI grade, no significant differences in discriminatory ability were found. Given its simplicity, ease of calculation, and availability from routine laboratory tests, the B/A ratio may serve as a practical biomarker for risk stratification in this patient population. These findings highlight its potential clinical utility in guiding prognosis and management strategies for patients with AD, although further validation in larger, multi-center studies is warranted.

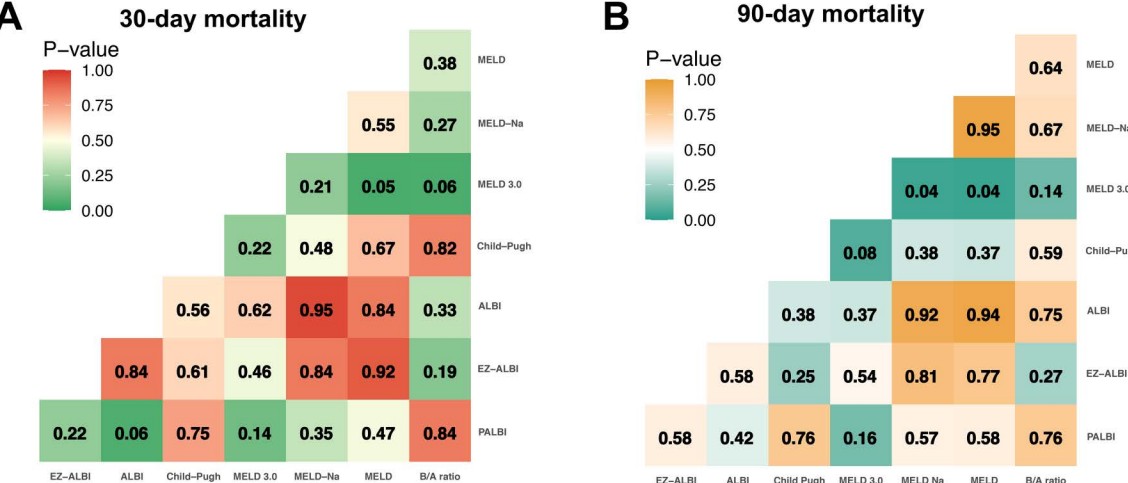

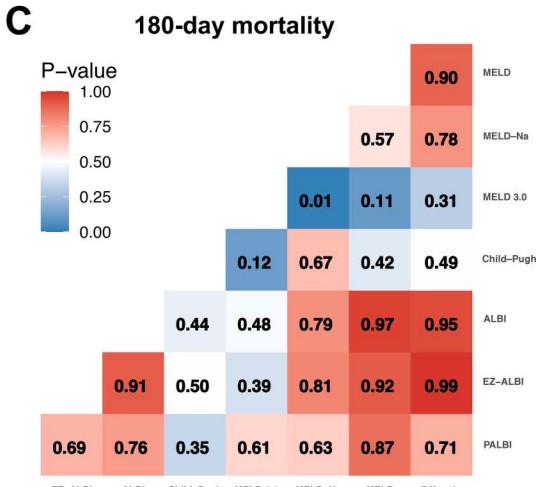

**Fig 6. Comparison of the prognostic value of the B/A ratio with established mortality prediction models and scoring systems.**

## The association between an increased B/A ratio and disease severity in patients with AD

The B/A ratio is derived from the bilirubin and albumin levels, and its elevation indicates either an increase in bilirubin, a decrease in albumin, or both occurring simultaneously. Elevated bilirubin is neurotoxic and contributes to bilirubin-induced neurological damage [28], which is a significant prognostic factor for mortality in patients with AD [29]. The relationship between elevated bilirubin and increased mortality has been well documented in experimental and clinical studies, reinforcing its role in overall mortality risk and acute hepatitis-related mortality [7,8,30,31].

Similarly, increasing evidence suggests that hypoalbuminemia is strongly associated with increased overall mortality [32–34]. Since albumin is synthesized in the liver, hypoalbuminemia is a direct consequence of hepatic dysfunction [9]. However, while hypoalbuminemia is linked to poor outcomes, the potential benefits of albumin supplementation remain inconclusive, as recent studies have produced mixed results [35,36].

Given the well-established roles of bilirubin and albumin in determining mortality risk, it is reasonable to infer that the B/A ratio also correlates with mortality, particularly in patients with liver disease. Our study reinforces this notion by

demonstrating that an elevated B/A ratio is independently associated with both short-term and long-term mortality in patients with AD. However, the prognostic significance of the B/A ratio appears to be mediated through its components, namely bilirubin and albumin, rather than through direct evidence supporting the ratio itself.

## Clinical implications and future research directions

Our study provides promising evidence supporting the prognostic role of the B/A ratio in AD. Additionally, prior studies have suggested a role for the B/A ratio in neonatal and pediatric populations [16–18,37]. Although the current body of evidence remains limited, these findings open avenues for future research to explore its broader applicability across different patient groups. Moreover, the B/A ratio has significant clinical utility, as it is a simple, readily available, and affordable biomarker that can be easily incorporated into routine clinical practice.

However, in our analysis, the independent prognostic value of the B/A ratio was no longer retained after adjusting for a full range of clinical variables, laboratory tests, and comorbid conditions, suggesting that its predictive capacity is influenced by these confounding factors. This underscores an important implication for clinical practice: the B/A ratio should not be used as a standalone indicator for prognosis or decision-making. Rather, it should be considered a supportive biomarker, used in conjunction with clinical assessments, laboratory findings, and established prognostic models to ensure a more accurate and comprehensive evaluation, ultimately improving patient care and outcomes. The B/A ratio may serve as an initial, readily obtainable marker for preliminary risk stratification. Once more detailed laboratory data and clinical information are available, clinicians can further refine prognostic evaluation using established models such as MELD, MELD-Na, MELD 3.0, or Child–Pugh. Alternatively, albumin–bilirubin–based indices, including ALBI, EZ-ALBI, and PALBI, may be applied according to the clinician's experience and preference, as these indices have demonstrated comparable prognostic performance in AD of cirrhosis, despite originally being developed for hepatocellular carcinoma prognosis [10–15,26].

## Limitations

Despite its strengths, including a prospective study design and recruitment from two centers, this study has several limitations. First, the study population is geographically restricted to Central Vietnam, which may limit the generalizability of the findings. Larger, multicenter studies involving more diverse populations are needed to validate these results. Additionally, the follow-up duration was limited to 180 days, and longer-term studies are necessary to assess the extended prognostic value of the B/A ratio. Future research should address these limitations by conducting large-scale, multiyear studies across broader geographic regions to confirm the clinical utility of this biomarker in liver disease management.

Another limitation is the lack of external validation in an independent cohort. However, we attempted to mitigate this by comparing the B/A ratio with well-established prognostic models, including the MELD and MELD 3.0 models. The results demonstrated that the B/A ratio had comparable or complementary prognostic value. Nonetheless, further studies are needed to validate its performance across different populations and clinical settings to ensure its reproducibility and generalizability.

Additionally, we did not assess ICU admission rates, hospital length of stay, or the development of organ failure as clinical outcomes. In Vietnam, cultural and religious beliefs significantly influence decisions regarding ICU admission, with many families opting for home-based end-of-life care rather than intensive treatment. This sociocultural factor introduces variability in ICU admission rates and hospital stays, making these parameters less reliable for assessing disease severity and prognosis. Consequently, we focused on all-cause mortality, which provides a more objective and comprehensive measure of patient outcomes, minimizing potential bias from external influences.

Finally, our study did not differentiate between specific causes of mortality but instead reported all-cause mortality. This approach was chosen because of challenges in accurately determining the primary cause of death, particularly in cases where patients were discharged home for end-of-life care, a common practice in Vietnam. Moreover, in critically ill

patients, multiple organ dysfunction often contributes to mortality, making it difficult to attribute death to a single cause. By reporting all-cause mortality, we aimed to provide a more holistic and unbiased assessment of patient outcomes while reducing the risk of misclassification bias.

## Conclusion

This study demonstrated that an increasing B/A ratio is associated with increased mortality in patients with AD. Notably, the B/A ratio serves as a practical and accessible prognostic marker, offering a straightforward and affordable approach to mortality risk assessment. Unlike complex scoring systems that require multiple parameters and intricate calculations, the B/A ratio can be easily derived from routine laboratory tests, making it highly applicable in diverse clinical settings. However, further large-scale, multicenter studies are needed to validate these findings and explore their broader clinical utility.

## Supporting information

**S1 Fig. Graphical abstract.**
(PDF)

**S1 Table. Baseline clinical characteristics.**
(DOCX)

## Acknowledgments

We would like to express our gratitude to Thu Chinh Tran Thi, Nam Van Duc Nguyen, Linh Duy Le, Minh Thai Le, Dieu Thuy Le Thi, Thu Hoa Nguyen Thi, Uyen Phuong Vuong Thi, Ngoc Nguyen Thi Bich, Hong Ngoc Nguyen Phan, Phuong Thao Ho, To Nguyen Thi for their valuable assistance in data collection and manuscript preparation.

## Author contributions

**Conceptualization:** Thang Viet Luong, Hai Nguyen Ngoc Dang.

**Data curation:** Hien Viet Quang Nguyen, Thang Viet Luong, Trung Nguyen Tran, Hai Nguyen Ngoc Dang.

**Formal analysis:** Hien Viet Quang Nguyen, Thang Viet Luong, Hai Nguyen Ngoc Dang.

**Investigation:** Hien Viet Quang Nguyen, Hai Nguyen Ngoc Dang.

**Methodology:** Thang Viet Luong, Hai Nguyen Ngoc Dang.

**Project administration:** Hien Viet Quang Nguyen, Thang Viet Luong, Hai Nguyen Ngoc Dang.

**Visualization:** Hai Nguyen Ngoc Dang.

**Writing – original draft:** Hien Viet Quang Nguyen, Thang Viet Luong, Trung Nguyen Tran, Hai Nguyen Ngoc Dang.

**Writing – review & editing:** Hien Viet Quang Nguyen, Thang Viet Luong, Trung Nguyen Tran, Hai Nguyen Ngoc Dang.

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
