## [Decision Letter · Decision Letter 0]

10 Jul 2025

Dear Dr. Nguyen Ngoc Dang,

Thank you for submitting your manuscript to PLOS ONE. After careful consideration, we feel that it may have some merit but does not currently  meet PLOS ONE’s publication criteria as it currently stands. Therefore, we invite you to submit a revised version of the manuscript that addresses the points raised during the review process.

AUC curves for MELD, MELD-Na and MELD 3.0 and Child-Pugh should be presented, as it seem that MELD 3.0 is almost superior to Bilirubin/Albumin, albeit your study was too small to identify a significant difference.

Without a power calculation to demonstrate that sample size was large enough to discriminate a difference in performance you cannot claim equivalence.

Explore if etiology of disease matter for survival

Patient likely need to divide in ACLF and non-ACLF, and explore at least in an explorative analysis.

For patients with ACLF the Clif score should be given.

publication criteria  and not, for example, on novelty or perceived impact.

We look forward to receiving your revised manuscript.

Kind regards,

Hans L. Tillmann

Academic Editor

PLOS ONE

Additional Editor Comments (if provided):

Reviewers' comments:

Reviewer's Responses to Questions

**Comments to the Author**

1. Is the manuscript technically sound, and do the data support the conclusions?

Reviewer #1: Partly

2. Has the statistical analysis been performed appropriately and rigorously?

Reviewer #1: Yes

3. Have the authors made all data underlying the findings in their manuscript fully available?

Reviewer #1: Yes

4. Is the manuscript presented in an intelligible fashion and written in standard English?

Reviewer #1: Yes

Reviewer #1: 1. Please give a proper rationale how B/A ratio will add to the already established prognostic scores.

2. Please explain whether it is superior/inferior to the existing scores.

3. Please revise the discussion and expand, adding the similar studies and comparing it with yours.

**Do you want your identity to be public for this peer review?** For information about this choice, including consent withdrawal, please see our Privacy Policy

Reviewer #1: No

---

## [Author Response · Author response to Decision Letter 1]

7 Aug 2025

REBUTTAL LETTER

Dear Editor and Reviewers,

We would like to express our sincere gratitude to the reviewers and the editor for their thorough review and insightful comments on our manuscript. Your valuable feedback has significantly helped us improve the quality and clarity of our work. We greatly appreciate your time and effort in providing constructive suggestions, and we are committed to addressing all the points raised to ensure that our manuscript meets the highest standards.

We would now like to respond to the specific comments and suggestions provided by the reviewers as follows:

RESPONSE TO EDITOR:

#1: AUC curves for MELD, MELD-Na and MELD 3.0 and Child-Pugh should be presented, as it seem that MELD 3.0 is almost superior to Bilirubin/Albumin, albeit your study was too small to identify a significant difference. Without a power calculation to demonstrate that sample size was large enough to discriminate a difference in performance you cannot claim equivalence.

Response:

We have addressed this point by including additional data in Supplementary Table 2, where the AUC values of the MELD, MELD-Na, MELD 3.0, and Child-Pugh scores for predicting 30-day, 90-day, and 180-day mortality are presented alongside the B/A ratio. Initially, the MELD-based models and Child-Pugh score exhibited slightly higher AUCs than the B/A ratio at the 30-day mortality endpoint. However, this advantage diminished over time, and the predictive performance of the B/A ratio became comparable at later timepoints.

Although our study sample is comparable to those of previous research in this field, we acknowledge that it may still be underpowered to detect statistically significant differences in discrimination performance across models. As you correctly pointed out, the absence of a formal power calculation limits our ability to claim equivalence between the B/A ratio and established prognostic scores. We have now clarified this limitation explicitly in the Discussion section. Thank you again for highlighting this important consideration.

#2: Explore if etiology of disease matter for survival

Response:

We are thankful for your valuable recommendation. In our Cox regression Model II, we have already adjusted for etiologies and precipitating factors, including key contributors to acute decompensation, as part of the fully adjusted model. To enhance clarity, we will elaborate on this aspect more explicitly in the Results section to ensure that readers fully understand the scope of adjustment.

We believe that conducting a separate analysis focusing solely on etiology-specific survival would be beyond the scope of our current study objectives and could potentially divert from the primary aim, which is to evaluate the prognostic utility of the B/A ratio in acute decompensation. Nevertheless, we appreciate your suggestion and consider it a valuable direction for future research.

#3: Patient likely need to divide in ACLF and non-ACLF, and explore at least in an explorative analysis.

Response:

We are thankful for your valuable recommendation. Initially, we did perform subgroup division between ACLF and non-ACLF patients (AD) and conducted comparative analyses. However, we initially decided not to include these data in detail to maintain a concise manuscript and avoid overburdening the reader with excessive content.

In light of your valuable suggestion, we have now reinstated these analyses in the Baseline Clinical Characteristics section and provided the detailed data in Supplementary Table 1. This approach ensures that readers who are specifically interested in the differences between ACLF and non-ACLF patients can access the relevant information, while preserving the overall clarity and focus of the main text. Thank you again for your helpful recommendation.

#4: For patients with ACLF the Clif score should be given.

Response:

We are thankful for your valuable recommendation. In response to your recommendation, we have included detailed information on relevant prognostic scores specifically for patients with ACLF, including CLIF-C ACLF, CLIF-SOFA, CLIF-C OF, Child-Pugh, MELD, MELD-Na, and MELD 3.0 in Supplementary Table 1. These additions provide a more comprehensive overview of disease severity and support further interpretation of outcomes in this subgroup. Thank you for highlighting this important point.

RESPONSE TO REVIEWER:

#1: Please give a proper rationale how B/A ratio will add to the already established prognostic scores.

Response:

We are thankful for your recommendation. Trong nghiên cứu của chúng tôi It is unsurprising that the bilirubin-to-albumin (B/A) ratio initially appeared independently associated with outcome after partial adjustment, yet lost statistical independence once all covariates were incorporated. Acute, critically ill populations are influenced by myriad interacting factors, and in real-world practice clinicians never rely on a single parameter to judge prognosis.

This rationale underpins the development of multi-component scores such as MELD, MELD-Na and MELD 3.0, which integrate several laboratory variables. The value of the B/A ratio lies in its simplicity: it offers a rapid, holistic snapshot while more complex calculations and additional tests are still pending. Our aim was therefore not to create a solitary marker that supplants existing tools, but to highlight a readily available index whose performance approaches that of more elaborate scores.

Clinical decision-making is analogous to diagnosis: no physician bases a diagnosis on one symptom or one test result. A meticulous examination is required, synthesising multiple clinical signs and investigations before reaching a final judgment. Likewise, we do not advocate replacing established clinical or laboratory assessments with the B/A ratio; rather, we propose it as a quick, initial risk-stratification aid in acute decompensated cirrhosis, prompting timely, decisive management while comprehensive scores are being calculated.

Accordingly, we deliberately use the qualifier “potential prognostic” in the title, and we now emphasise in the Discussion that the B/A ratio is intended to complement—not replace—other critical clinical and paraclinical information.

#2: Please explain whether it is superior/inferior to the existing scores.

Response:

As previously mentioned, our study did not aim to demonstrate the superiority of the B/A ratio over established prognostic models. We fully acknowledge that such a comparison would be inappropriate, as multi-component scores like MELD, MELD-Na, and MELD 3.0 incorporate a broader range of laboratory parameters and clinical information, which naturally enhances their prognostic accuracy.

Instead, our objective was to highlight the practical value of the B/A ratio—its simplicity, accessibility, and ease of use—as a rapid, integrative marker that can provide early prognostic insight while awaiting more comprehensive test results. While the B/A ratio may not outperform these established models, its comparable performance and clinical convenience make it a valuable adjunctive tool, particularly in resource-limited settings or when rapid decision-making is essential.

#3: Please revise the discussion and expand, adding the similar studies and comparing it with yours.

Response:

Initially, we intended to include comparisons with similar studies; however, the majority of existing literature on the bilirubin-to-albumin (B/A) ratio focuses on neonatal or pediatric populations. Due to fundamental physiological and clinical differences between children and adults, especially in liver disease progression and systemic responses, we believe that direct comparisons across age groups would be inappropriate and potentially misleading.

Therefore, rather than drawing comparisons with these pediatric-focused studies, we opted to emphasize the pathophysiological rationale behind the prognostic value of the B/A ratio in adults with acute decompensation. This approach, in our view, aligns more closely with the aims of our research and provides a more relevant context for interpreting our findings. Nonetheless, we fully agree that comparative discussion with future adult-focused studies will be essential as more evidence emerges.

In addition to the revisions outlined above, we have diligently addressed all reviewer and editor comments. We’ve also conducted a thorough review of the manuscript for grammatical precision, scientific terminology, reference accuracy, and unit consistency, adhering to journal guidelines. These efforts have significantly elevated the manuscript’s quality and clarity. We eagerly await your feedback on this revised version.

Thank you once again for your invaluable contributions.

Sincerely,

The Authors

---

## [Editor Report · Decision Letter 1]

22 Aug 2025

Dear Dr. Nguyen Ngoc Dang,

Thank you for submitting your manuscript to PLOS ONE. After careful consideration, we feel that it has merit but does not fully meet PLOS ONE’s publication criteria as it currently stands. Therefore, we invite you to submit a revised version of the manuscript that addresses the points raised during the review process.

As you may not be familiar with prior scores related to Albumin and Bilirubin, different from MELD I now listed those score below for the authors for easy incorporation; I also including the references are listed  below.

There are previous scores using Albumin and Bilirubin one including Platelets assessing the “liver reserve”. While this is in the setting of HCC resection, especially ACLF is where liver reserve plays a role, and thus should be included in assessment.

<table border="0" cellpadding="0"> <thead> <tr> <th>

**Noninvasive Liver Reserve Models**

</th> <th>

**Formula**

</th> </tr> </thead> <tbody> <tr> <td>

BiLE Score[1]

6.9 threshold

</td> <td>

Bilirubin (μmol/L)/100 + lactate (mmol/L)

+ 4 (in case of indeterminate ALF, Budd-Chiari syndrome, or phenprocoumon toxicity)

-2 (in case of acetaminophen toxicity)

+0 (in case of any other ALF etiology

</td> </tr> <tr> <td>

ALBI grade[2]

 1/2/3

(≤−2.6/>−2.6 and ≤−1.39/>−1.39)

</td> <td>

(log_10_ (Bilirubin (μmol/L)) × 0.66) + (Albumin (g/L) × (−0.085))

</td> </tr> <tr> <td>

EZ-ALBI grade[3]

 1/2/3

(≤−34.4, >−34.4 and ≤−22.2, >−22.2)

</td> <td>

Bilirubin (mg/dL) − (9 × Albumin (g/dL))

</td> </tr> <tr> <td>

PALBI grade[4]

 1/2/3

(≤−2.53), score > −2.53 and ≤−2.09)/(score > −2.09)

</td> <td>

2.02 × log_10_ Bilirubin (μmol/L) level − 0.37 × (log_10_ Bilirubin level)^2^ − 0.04 × Albumin level − 3.48 × log_10_ Platelet count (1000/μL) + 1.01 × (log_10_ Platelet count)^2^

</td> </tr> <tr> <td>

PAL grade 1/2/3

(score ≤ −3.77, score > −3.77 and ≤−3.04,

score > −3.04

</td> <td>

0.777 × Albumin (g/dL) − 0.575 × (log_10_ Platelet count) (10^4^ /μL)

</td> </tr> </tbody></table>

It would be important to demonstrate that a new score is either substantially simpler or substantially more accurate in predicting mortality.

Of the above models, you can potentially skip the 1^st^ one, as it does not include Albumin, but the others would need to be explore in comparison to the new score.

While the study may be too small, a power calculation, to indicate an ideal study size would still be required.

[1] Hadem J, Stiefel P, Bahr MJ, Tillmann HL, Rifai K, Klempnauer J, Wedemeyer H, Manns MP, Schneider AS. Prognostic implications of lactate, bilirubin, and etiology in German patients with acute liver failure. Clin Gastroenterol Hepatol. 2008 Mar;6(3):339-45. doi: 10.1016/j.cgh.2007.12.039. PMID: 18328438

[2] Johnson PJ, Berhane S, Kagebayashi C, Satomura S, Teng M, Reeves HL, O'Beirne J, Fox R, Skowronska A, Palmer D, Yeo W, Mo F, Lai P, Iñarrairaegui M, Chan SL, Sangro B, Miksad R, Tada T, Kumada T, Toyoda H. Assessment of liver function in patients with hepatocellular carcinoma: a new evidence-based approach-the ALBI grade. J Clin Oncol. 2015 Feb 20;33(6):550-8. doi: 10.1200/JCO.2014.57.9151. Epub 2014 Dec 15. PMID: 25512453; PMCID: PMC4322258.

[3] Kariyama K, Nouso K, Hiraoka A, Wakuta A, Oonishi A, Kuzuya T, Toyoda H, Tada T, Tsuji K, Itobayashi E, Ishikawa T, Takaguchi K, Tsutsui A, Shimada N, Kudo M, Kumada T. EZ-ALBI Score for Predicting Hepatocellular Carcinoma Prognosis. Liver Cancer. 2020 Dec;9(6):734-743. doi: 10.1159/000508971. Epub 2020 Oct 23. PMID: 33442542; PMCID: PMC7768152.

[4]Roayaie S, Jibara G, Berhane S,  Tabrizian P, Park JW, Jijin Yang J, Yan L, Guohong Han G, Izzo F,  Chen M,  Blanc JF, Kudo M, Roberts LR. Sherman M, Johnson P. PALBI-An Objective Score Based on Platelets, Albumin & Bilirubin Stratifies HCC Patients Undergoing Resection &Ablation Better than Child’s Classification. Poster Sessions. Hepatology. 2015;62 Suppl 1:631A

We look forward to receiving your revised manuscript.

Kind regards,

Hans L. Tillmann

Academic Editor

PLOS ONE
---

## [Author Response · Author response to Decision Letter 2]

15 Sep 2025

RESPONSE LETTER

Dear Editor and Reviewers,

We greatly appreciate your time and effort in providing constructive suggestions, and we are committed to addressing all the points raised to ensure that our manuscript meets the highest standards.

We would now like to respond to the specific comments and suggestions provided by the reviewers as follows:

RESPONSE TO EDITOR AND REVIEWER:

#1: There are previous scores using Albumin and Bilirubin one including Platelets assessing the “liver reserve” such as ALBI, EZALBI, PALBI. While this is in the setting of HCC resection, especially ACLF is where liver reserve plays a role, and thus should be included in assessment. It would be important to demonstrate that a new score is either substantially simpler or substantially more accurate in predicting mortality. Of the above models, you can potentially skip the 1st one, as it does not include Albumin, but the others would need to be explore in comparison to the new score.

Response:

We sincerely appreciate this insightful comment. In response, we have incorporated the suggested scores (ALBI, EZ-ALBI, and PALBI) into our analysis, as presented in Supplementary Table 3. The results show that the B/A ratio did not differ significantly from these established models in terms of predictive performance. This further supports the notion that the B/A ratio, despite its simplicity, achieves comparable prognostic accuracy to more complex scoring systems.

#2: While the study may be too small, a power calculation, to indicate an ideal study size would still be required.

Response:

In accordance with the suggestion, we have added a power calculation to the Methods section to indicate the ideal study size.

Thank you once again for your invaluable contributions.

Sincerely,

The Authors

---

## [Editor Report · Decision Letter 2]

8 Oct 2025

Dear Dr. Nguyen Ngoc Dang,

Thank you for submitting your manuscript to PLOS ONE. After careful consideration, we feel that it has merit but does not fully meet PLOS ONE’s publication criteria as it currently stands. Therefore, we invite you to submit a revised version of the manuscript that addresses the points raised during the review process.

We look forward to receiving your revised manuscript.

Kind regards,

Hans L. Tillmann

Academic Editor

PLOS ONE
---

## [Author Response · Author response to Decision Letter 3]

26 Oct 2025

RESPONSE LETTER

Dear Editor,

We greatly appreciate your time and effort in providing constructive suggestions, and we are committed to addressing all the points raised to ensure that our manuscript meets the journal’s standards.

We respond to each of your comments as follows:

RESPONSE TO EDITOR:

#1: When exploring the role to Albumin to Bilirubin, you need to cite in the introduction the previously and more established model, do NOT Push them into a supplement. The key question for you paper is if your new score is relevantly differnt from established scores that were already subsequently externally validated.

Response:

Thank you for this valuable suggestion. We have revised the introduction section to include citations of established models and previous. In addition, we have refined the introduction section to clearly define the aim of our study, explicitly stating that it compares this ratio with previously validated scoring systems.

#2: You score does not need to be superior to previous score to make you paper acceptable, but it should be more clarly demonstarted how you score compared to other scores performs.

Response:

We appreciate your clarification. Accordingly, we have expanded the discussion section to provide a clearer comparison between the bilirubin-to-albumin ratio and existing prognostic models.

#3: Please also better explain what your saple size calculation is based on. I believe your study did not aim to assess prevalence of cirrhosis, but prediction of mortality related to bilirubin to albumin ratio.

Response:

Thank you for this insightful comment. We have revised the method section to explain our sample size calculation. As there were no prior studies directly evaluating this ratio in our target population, we referenced recent related studies to establish a reasonable estimate for sample size determination.

Once again, we are deeply grateful for your constructive feedback and the opportunity to further refine our manuscript. Your comments have greatly contributed to strengthening the scientific rigor and presentation of our work. We sincerely hope that the revisions meet your expectations.

With our best regards,

The Authors

---

## [Editor Report · Decision Letter 3]

29 Oct 2025

Dear Dr. Nguyen Ngoc Dang,

Thank you for submitting your manuscript to PLOS ONE. After careful consideration, we feel that it has merit but does not fully meet PLOS ONE’s publication criteria as it currently stands. Therefore, we invite you to submit a revised version of the manuscript that addresses the points raised during the review process.

We look forward to receiving your revised manuscript.

Kind regards,

Hans L. Tillmann

Academic Editor

PLOS ONE

Journal Requirements:

Additional Editor Comments :

You getting there.

The revision, did not show well how your new model compares the the established models, this would be crucial and needs to be in the main manuscript, please demonstrate how your model performs to established models, thus your supplemental table 2 & 3 needs to be part of the man manuscript, you should also include the information from supplemental table 2 & 3 into the abstract.

For table 2 you may also want to show the p-value comparing your A/B score to MELD etc, as in supplemental table 3.

You may also want to point out, that you show that you demonstrate that ALBI score was not significantly better than your more simple score, that can be calculated by hand vs. ALBI score, most of us would need a calculator. Though you would admit that the EZ-ALBI score can be hand calculated.

But you can point out also, that while it was developed for HCC, you show that is also works for de-compensated cirrhosis independent of HCC.

---

## [Author Response · Author response to Decision Letter 4]

2 Nov 2025

RESPONSE LETTER

Dear Editor,

We greatly appreciate your time and effort in providing constructive suggestions, and we are committed to addressing all the points raised to ensure that our manuscript meets the journal’s standards.

We respond to each of your comments as follows:

RESPONSE TO EDITOR:

#1: Your supplemental table 2 & 3 needs to be part of the man manuscript, you should also include the information from supplemental table 2 & 3 into the abstract. For table 2 you may also want to show the p-value comparing your A/B score to MELD etc, as in supplemental table 3.

Response:

Thank you very much for your comment. Initially, we were concerned that including too many tables and figures in the main text might increase the total page count. However, following your suggestion, we have incorporated the data from Supplemental Tables 2 and 3 into Table 3 of the main manuscript. In addition, the comparative AUC values between our index and established prognostic models have been added to Figure 6 for clearer visual presentation. We also updated the Abstract to include relevant information from these tables, while ensuring that the word limit requirements were not exceeded.

#2: You may also want to point out, that you show that you demonstrate that ALBI score was not significantly better than your more simple score, that can be calculated by hand vs. ALBI score, most of us would need a calculator. Though you would admit that the EZ-ALBI score can be hand calculated. But you can point out also, that while it was developed for HCC, you show that is also works for de-compensated cirrhosis independent of HCC.

Response:

Thank you very much for your comment. We have revised the Discussion section, specifically the part on clinical implications, to emphasize that previously developed indices can also be applied in this context based on the data from our study.

JOURNAL REQUIREMENTS:

Response: We have carefully reviewed the references suggested by the reviewers and editors, and only cited those that are directly relevant to our study. Unrelated references were not included. In addition, we thoroughly rechecked our reference list to ensure accuracy and to confirm that none of the cited papers have been retracted. No retracted publications are cited in our revised manuscript.

Once again, we are deeply grateful for your constructive feedback and the opportunity to further refine our manuscript. Your comments have greatly contributed to strengthening the scientific rigor and presentation of our work. We sincerely hope that the revisions meet your expectations.

With our best regards,

The Authors

---

## [Editor Report · Decision Letter 4]

6 Nov 2025

The bilirubin-to-albumin ratio as a potential prognostic biomarker for all-cause mortality in patients with acute decompensation of cirrhosis: A prospective study

PONE-D-25-26872R4

Dear Dr. Nguyen Ngoc Dang,

We’re pleased to inform you that your manuscript has been judged scientifically suitable for publication and will be formally accepted for publication once it meets all outstanding technical requirements.

Kind regards,

Hans L. Tillmann

Academic Editor

PLOS ONE
---

## [Editor Report · Acceptance letter]

PONE-D-25-26872R4

PLOS One

Dear Dr. Nguyen Ngoc Dang,

I'm pleased to inform you that your manuscript has been deemed suitable for publication in PLOS One. Congratulations! Your manuscript is now being handed over to our production team.

Kind regards,

on behalf of

Dr. Hans L. Tillmann

Academic Editor

PLOS One